# Vunakizumab-IL22, a Novel Fusion Protein, Promotes Intestinal Epithelial Repair and Protects against Gut Injury Induced by the Influenza Virus

**DOI:** 10.3390/biomedicines11041160

**Published:** 2023-04-12

**Authors:** Chenchen Shi, Chang Su, Lifeng Cen, Lei Han, Jianguo Tang, Zetian Wang, Xunlong Shi, Dianwen Ju, Yiou Cao, Haiyan Zhu

**Affiliations:** 1Department of Biological Medicines & Shanghai Engineering Research Center of ImmunoTherapeutics, School of Pharmacy, Fudan University, Shanghai 201203, China; 16111030031@fudan.edu.cn (C.S.); lifengcen007@163.com (L.C.); 17111030056@fudan.edu.cn (L.H.); xunlongshi@fudan.edu.cn (X.S.); dianwenju@fudan.edu.cn (D.J.); 2Division of Spine, Department of Orthopedics, Tongji Hospital, School of Medicine, Tongji University, Shanghai 200092, China; 3Department of Surgery, Minhang Hospital, Fudan University, Shanghai 201100, China; suchang_jx@126.com; 4Key Laboratory of Whole-Period Monitoring and Precise Intervention of Digestive Cancer (SMHC), Minhang Hospital & AHS, Fudan University, Shanghai 201100, China; 5Department of Trauma-Emergency & Critical Care Medicine, Shanghai Fifth People’s Hospital, Fudan University, Shanghai 200240, China; tangjianguo@5thhospital.com (J.T.); 18321127738@163.com (Z.W.)

**Keywords:** influenza A virus, Vunakizumab-IL22, intestinal barrier, gut–lung axis, dextran sulfate sodium

## Abstract

Secondary immune damage to the intestinal mucosa due to an influenza virus infection has gained the attention of investigators. The protection of the intestinal barrier is an effective means of improving the survival rate in cases of severe pneumonia. We developed a fusion protein, Vunakizumab-IL22(vmab-IL22), by combining an anti-IL17A antibody with IL22. Our previous study showed that Vunakizumab-IL22 repairs the pulmonary epithelial barrier in influenza virus-infected mice. In this study, we investigated the protective effects against enteritis given its anti-inflammatory and tissue repair functions. The number of goblet cells and the expression of zonula occludens protein 1(ZO-1), Mucin-2, Ki67 and IL-22R were determined by immunohistochemistry (IHC) and quantitative RT-PCR in influenza A virus (H1N1)-infected mice. The expression of NOD-like receptor pyrin domain containing 3 (NLRP3) and toll- like-receptor-4 (TLR4) was assayed by IHC in the lungs and intestine in HIN1 virus-induced mice to evaluate the whole efficacy of the protective effects on lungs and intestines. Consequently, Cytochrome C, phosphorylation of nuclear factor NF-kappaB (p-NF-κB), IL-1β, NLRP3 and Caspase 3 were assayed by Western blotting in dextran sulfate sodium salt (DSS)-treated mice. Treatment with Vunakizumab-IL22 improved the shortened colon length, macroscopic and microscopic morphology of the small intestine (*p* < 0.001) significantly, and strengthened the tight junction proteins, which was accompanied with the upregulated expression of IL22R. Meanwhile, Vunakizumab-mIL22 inhibited the expression of inflammation-related protein in a mouse model of enteritis induced by H1N1 and DSS. These findings provide new evidence for the treatment strategy for severe viral pneumonia involved in gut barrier protection. The results suggest that Vunakizumab-IL22 is a promising biopharmaceutical drug and is a candidate for the treatment of direct and indirect intestinal injuries, including those induced by the influenza virus and DSS.

## 1. Introduction

Influenza is a global public health challenge due to its high morbidity [1,2]. The therapeutic effects of vaccines and antiviral drugs are limited due to viral mutations and missed windows for optimal treatment. The host response, including immune modulators, could offer more efficient therapeutic strategies for infectious diseases. Intestinal immune injury is a common extra-pulmonary complication of respiratory virus infections. Severe pneumonia causes acute respiratory distress syndrome and even systemic inflammation, which is usually accompanied by the increased permeability of the intestinal barrier and gut microbe imbalances included in the gut–lung axis [3,4,5,6,7,8,9]. Intestinal barrier function plays a prognostic role in predicting the progression to severe respiratory failure. Repairing the intestinal barrier would be important and helpful for preventing pathogenic bacteria from reaching other parts of the body and causing secondary complications.

Previous studies, including some by our group, have confirmed that an influenza A virus (IAV) infection leads to gut dysbiosis, intestinal barrier damage and immune function disorder [3,4,5,6]. The cross-talk between the gut and lungs is mediated by Th17 cells, which contribute to the pathogenesis of gut and lung injury through their differentiation and migration across the gut–lung axis [7,8]. A recent study showed that fecal microbe transplantation can restore the Th17/Treg balance and attenuate inflammation and lung injury in mice with pneumonia [9,10]. This evidence suggests that the restoration of intestinal homeostasis, including microbiota and intestinal barrier repair, is an effective strategy for treating lung injury.

Interleukin 17 (IL17) is considered an important inflammatory mediator in inflammatory immune injury. There are multiple sources of IL17, but IL17 derived from Th17 cells has been implicated in inflammatory lung diseases [11]. IL17A is also closely associated with intestinal disease in mice and humans, but conflicting evidence has drawn the role of 17 in the gut into question. In a viral pneumonia model, IL17 has been demonstrated to mediate lung injury [7]. In contrast, studies have also shown that IL17 exerts multiple protective effects, such as antimicrobial, antiproliferative, and antiapoptotic effects; tissue damage prevention; and enhanced epithelial repair [12,13]. Anti-IL17 antibodies can protect against the pulmonary inflammatory injury induced by lipopolysaccharides and the H1N1 virus, and these effects are primarily mediated by the inhibition of inflammatory cytokines [13]. The effect of blocking IL17 on intestinal injury in viral pneumonia is unknown.

Interleukin 22 (IL22) is well-known as a protective cytokine that helps in maintaining the pulmonary and intestinal epithelial barrier. On binding to its receptor, IL22 primarily activates the signal transducer and activator of transcription 3 (STAT3) through the Janus kinase (JAK)–STAT pathway [14]. This IL22-mediated signaling pathway promotes local tissue regeneration and host defense and the expression of anti-apoptotic genes [15]. IL22 is essential for repairing lung tissue and maintaining epithelial cell homeostasis after an influenza virus infection, suggesting its promising role as an immunotherapy agent for interstitial lung disease [16,17]. Meanwhile, a deficiency of IL22 can cause a delay in barrier repair and the exacerbation of inflammatory pathology [18]. These results suggest that IL22 acts as an important regulator for maintaining intestinal homeostasis, exerting antimicrobial effects, promoting proliferation, and inhibiting apoptosis to prevent intestinal injury [19,20,21]. A recent study showed that insufficient levels of IL22 could be responsible for aberrant epithelial repair and adverse immune responses, leading to increased replication of the influenza virus and severe pneumonia [22].

The protective effect of recombinant IL22 on intestinal barrier damage in inflammatory bowel disease has been confirmed. The complexity of the cytokine network limits the clinical application of recombinant proteins. IL22 can easily lose its protective effect and even become pathogenic under a high-IL17 microenvironment [23,24,25]. In view of the function of IL17 and IL22 in inflammatory disease, a novel bifunctional fusion protein was successfully constructed to block IL17A and increase IL22 in order to achieve the dual effects of inflammation reduction and repair. This fusion protein was found to significantly alleviate lung injury and effectively promote the repair of the alveolar epithelium [26]. Blocking IL17A contributes to the inhibition of pulmonary inflammation and also enhances the alveolar epithelial repair efficacy of extraneous IL22 to achieve the two functions. However, its protective effect against immune enteritis is unknown.

This study focused on evaluating the protective effects of the bifunctional protein against intestinal injury in a viral pneumonia model and comparing them with those of the anti-IL17A antibody and recombinant IL22 alone or in combination. The goal was to provide a basis for the application of this bifunctional protein in the treatment of immune enteritis induced by IAV. Further, the results were subsequently validated in a direct injury enteritis model. The findings would demonstrate that the protection of the intestinal barrier is helpful for reducing lung injury, and indicate that the recombinant protein could alleviate the multiple organ injury induced by severe influenza virus infection. Meanwhile, this study provides experimental evidence for the potential use of the fusion protein for the treatment intestinal barrier damage induced by direct and indirect cause.

## 2. Materials and Methods

### 2.1. Preparation of Vunakizumab-IL22 (vmab-mIL22)

Details of the construction, expression, and purification of this fusion protein are provided in our previous paper [26]. In brief, the variable regions of the heavy chain and light chain of a mouse anti-IL17A antibody (vunakizumab, vmab) were synthesized and cloned separately into an expression vector. The mouse IL22 sequence was inserted at the C terminus of the vmab heavy chain to construct the plasmid pTT5-vmab-mIL22. The fusion proteins were produced in CHO cells via liposome-mediated transient gene expression. The protein of interest was harvested and purified using size-exclusion chromatography and protein affinity chromatography.

### 2.2. Influenza Virus 

A mouse-adapted strain of IAV (A/FM/1/47; H1N1) was supplied by the Shanghai Center for Disease Control and Prevention (Shanghai, China) and maintained at the center for anti-inflammation and anti-virus drug screening (School of Pharmacy, Fudan University, Shanghai, China). The virus could multiply in the lungs of mice and was stored at −80 °C before use. Using survival tests in mice, the median lethal dose (LD_50_) was determined to be a 5 × 10^−4.3^ dilution of the storage solution.

### 2.3. Animals 

Specific pathogen-free (SPF) male BALB/c mice (4–6 weeks, 14–16 g, SPF II, Certificate No. SCXK 2013-0018) and C57BL/6 mice (6–8 weeks, SPF II, Certificate No. SCXK 2013-0018) were obtained from the Shanghai Slaccas Company (Shanghai, China). Animals were housed in independent ventilated cages (IVCs) with access to feed and water under a 12 h light/dark cycle. All protocols used in this study were approved by the Animals Ethical Committee of the School of Pharmacy, Fudan University (Approval No. 2021-09-SY-ZHY-96). 

#### 2.3.1. Experimental Enteritis Model Induced by H1N1 Infection

BALB/c mice were intranasally inoculated with 30 μL Dulbecco’s Modified Eagle Medium (DMEM) and served as the normal control group. Thirty mice were weighed, anesthetized with isoflurane, and then intranasally infected with a 30 μL H1N1 dilution at a dose of 2LD_50_ to obtain an H1N1 infection model. The infected mice were randomly divided into five groups (*n* = 6), as follows: H1N1 + vehicle (PBS); H1N1 + 8.58 mg/kg vmab; H1N1 + 5 mg/kg mIL22Fc; H1N1 + 10.66 mg/kg vmab-mIL22; and H1N1 + 8.58 mg/kg vmab + 5 mg/kg mIL22Fc (combination). The doses of vmab and vmab-mIL22 were equimolar to that of mIL22Fc. The treatment was administered twice intravenously during the 5-day observation period. The administration time points were 2.5 h and 48 h after infection. The mice were sacrificed on the 5th day. The upper lobe of the right lung and 1 cm of the proximal small intestine were fixed in 4% formalin for pathological assessment. The remaining portions of the lung and small intestine, colon, liver, and spleen were cryopreserved for subsequent analyses. 

#### 2.3.2. Experimental Enteritis Model Induced by Dextran Sodium Sulfate (DSS) 

To further evaluate the protective efficacy of the fusion protein against direct gut injury caused by a non-immune factor, a DSS-induced ulcerative colitis (UC) model was employed. All C57BL/6 mice were randomly divided into six groups (*n* = 8). The experimental group and therapeutic doses were the same as those described in Section 2.3.1. The control group, however, was allowed free intake of distilled water and received 0.5% sodium carboxymethyl cellulose (CMC Na) once a day for 7 days. The DSS group received free intake of 2.5% DSS (MP Biochemicals, CA, USA) and was treated with 0.5% CMC Na for 7 days. The vmab, mIL22Fc, vmab-mIL22 and vmab/mIL22 combination groups received free intake of 2.5% DSS for 7 days, followed by intravenous administration of 8.58 mg/kg vmab, 5 mg/kg mIL22Fc, 10.66 mg/kg vmab-mIL22 and 8.58 mg/kg vmab + 5 mg/kg mIL22Fc (combination), respectively. During the 7-day observation period, the treatment was administered intravenously twice, first on day 1 and then on day 4. The body weights of mice were measured every day. Mice were sacrificed on day 7. A 1 cm portion of the colon was fixed in 4% formalin for pathological assessment, and the remaining colon tissue was stored at −80 °C for subsequent analyses. 

### 2.4. Pathological Observation and Periodic Acid-Schiff Stain (PAS)

The lung and small intestine tissues were stored in 4% paraformaldehyde for 24 h. After dehydration and embedding in paraffin, the tissues were cut into 4-μm-thick sections. Following the standard process, the sections were stained with hematoxylin and eosin (H&E) and the Alcian Blue/Periodic acid-Schiff (AB-PAS) (BA-4080A, Baso Biotechnology, Wuhan, China) stain to observe the pathological changes and the number of goblet cells. Histological images were acquired using an Olympus SLIDEVIEW VS200 microscope (Olympus Corporation, Tokyo, Japan). 

### 2.5. Enzyme-Linked Immunosorbent Assay (ELISA)

Intestinal tissue homogenates (100 mg tissue/mL PBS) from six groups of mice were prepared using Tissue Grinder (Jingxin Pharmaceutical Co., Ltd., Shanghai, China) at 50 Hz for 1 min. The protein quantity was detected using a BCA Kit (P0012S, Beyotime Biotechnology, Shanghai, China). IL22 expression in the intestinal homogenates was measured on day 5 after H1N1 infection using IL22 ELISA kits (Abclone, Wuhan, China). All assays were performed according to the manufacturer’s instructions. 

### 2.6. Immunohistochemistry (IHC) Assay

Formalin-fixed, paraffin-embedded 4 μm sections of lungs and intestinal tissue from H1N1-infected and DSS-treated mice were used for IHC. After deparaffinization with xylene, three sections from each group were incubated in a citrate buffer solution for antigen retrieval (20 min). Subsequently, the tissues were blocked with 5% bovine serum albumin for 30 min and then incubated at 4 °C for 24 h with 50 μL of primary antibodies against zonula occludens protein 1 (ZO-1, ab96587, abcam, Cambridge, UK); Mucin-2 (MUC2, ab76774, abcam, Cambridge, UK); Ki67 (ab15580, abcam, Cambridge, UK); toll-like receptor 4 (TLR4, ab22048, abcam, Cambridge, UK); and NOD-like receptor thermal protein domain associated protein 3 (NLRP3, ab214185, abcam, Cambridge, UK) at a 1:100 dilution. The following day, sections were incubated with the secondary antibody, goat anti-rabbit IgG (A0208, Beyotime, Shanghai, China), at 1:300 for 1 h at room temperature 22 °C. Finally, the samples were sealed with neutral balsam, and images were acquired using an Olympus SLIDEVIEW VS200 microscope (Olympus Corporation, Tokyo, Japan). 

### 2.7. Quantitative Real-Time PCR (RT-qPCR) Analysis

Tissue from the right upper lung lobe, small intestine, colon, liver and spleen (100 mg) of each mouse was obtained on day 4 and placed in 1 mL of Trizol (15596018, Invitrogen, Carlsbad, CA, USA) for extracting the total RNA. Total RNA was treated with DNAse-free Kit to remove genomic DNA contamination, and was used for reverse transcription reaction with RT reagent Kit (RR037A, Takara, Kusatsu, Japan) to obtain cDNA template. The 20 μL RT-qPCR reaction mix was prepared with 2 μL of template DNA, 0.4 μL of specific primers,10 μL of TB Green Premix Ex Taq (RR420A, Takara, Kusatsu, Japan), and 7.2 μL of water. The PCR amplification procedure was as follows: pre-denaturation at 95 °C for 30 s and 40 cycles of denaturation at 95 °C for 5 s, annealing at 60 °C, and extension for 30 s. RT-qPCR experiments were performed using six animals from each group. Specific primers against glyceraldehyde-3-phosphate dehydrogenase (GAPDH) and IL22R are listed in Table 1. The procedure was run on a Veriti 96-Well Thermal Cycler (Applied Biosystems, Bedford, MA, USA). The mRNA levels of IL22R were normalized to those of GAPDH and were expressed as a fold increase over the mean gene expression level in control mice. 

### 2.8. Western Blotting

Protein was extracted from the proximal 1 cm portion of the colon tissue using RIPA lysis buffer containing protease inhibitor and quantified using a BCA protein assay kit (Beyotime, Shanghai, China). Then, 20 μg protein was separated using sodium dodecyl sulfate-polyacrylamide gel electrophoresis and transferred to a polyvinylidene fluoride membrane. After blocking for 1 h, the membranes were incubated with primary antibodies such as anti-Cytochrome C antibody (ab133504, abcam, Cambridge, UK); anti-Caspase 3 antibody (ab184787, abcam, Cambridge, UK); anti-Cleaved Caspase 3 antibody (ab214430, abcam, Cambridge, UK); anti-NLRP3 antibody (ab270449, abcam, Cambridge, UK); anti-Cleaved-IL-1ββ antibody (mAb#63124, CST, Boston, MA, USA); anti-phospho NF-kB p65 (Ser536) antibody (mAb#3033, CST, Boston, MA, USA); anti-phospho IκBα (Ser32) antibody (mAb#2859, CST, Boston, MA, USA); and anti-GAPDH antibody (AF0006, Beyotime, Shanghai, China) overnight at room temperature. Then, the membranes were incubated with the HRP-conjugated secondary antibody for 1 h. The color was developed with an electrochemiluminescence (ECL) reagent (Beyotime Biotechnology, Shanghai, China) and images were captured using the Bio-Rad gel imager (Bio-Rad, Hercules, CA, USA). The test method was based on previous papers from our research group [26].

### 2.9. Data Analysis

GraphPad Prism 7 Software was used for all statistical analyses. Data were expressed as the mean ± standard, and differences among variables were evaluated using a one-way ANOVA followed by Dunnett’s test. For paired group comparisons, we used Student’s *t*-test. ## *p* < 0.01 for comparison of mice in H1N1 group to the normal group, *** *p* < 0.001, * *p* < 0.05 for comparison of mice in treated group to the H1N1 group.

## 3. Results

### 3.1. Vmab-IL22 Protects against Lung and Intestinal Injury Induced by Influenza Virus Infection

IAV infection leads to direct lung injury and indirect intestinal injury via the gut–lung axis [7,8]. Our previous study demonstrated that vmab-IL22 protects against lung injury induced by IAV. In this study, compared with the normal group, the infected mice showed a shorter colon length (*p* < 0.001) (Figure 1A, B). H&E-stained sections revealed the damage to villous morphology and crypt cells in the small intestine and the elevated infiltration of inflammatory cells in the lamina propria and lung in H1N1-infected mice (Figure 1C). The administration of vmab-IL22 significantly reduced the lung injury and alleviated the reduction in colon length (*p* < 0.001) (Figure 1A–C). Compared with vmab and mIL22 Fc alone, vmab-IL22 exerted greater protective effects against lung and intestinal injury, as indicated by the relieved pathological states (Figure 1B). 

### 3.2. Vmab-IL22 Strengthens the Intestinal Barrier in Influenza Virus-Infected Mice

An intact intestinal epithelial barrier is vital for preserving the integrity of the host digestive system and preventing disease development. For IHC analysis, tissues from the small intestine were fixed in formalin and embedded in paraffin blocks. Sections were stained with PAS stain, and IHC was performed for MUC2, ZO-1, and Ki67 to assess the number of regenerating crypts and the mitotic index. As shown in (Figure 2A–C), compared with the normal control group, the expression of the tight junction proteins MUC2 and ZO-1 and the number of mucous-secreting cells were lower in the small intestines of H1N1-infected mice, indicating that the mucin layer structure in the intestinal barrier was damaged. The damage to the intestinal barrier was significantly inhibited following vmab-mIL22 treatment. Ki67, as an epithelial proliferation marker, is indispensable for cell proliferation. IHC analysis of the proliferative marker Ki67 revealed a significant increase in expression in intestines of H1N1-infected mice (Figure 2D). The treatment with vmab-mIL22 inhibited the expression of Ki67, indicating the protective efficacy of vmab-mIL22 is through inhibiting pathological repair. 

### 3.3. Vmab-IL22 Promotes the Expression of IL22R in the Small Intestine in Influenza Virus-Infected Mice

To determine the target organ of the vmab-IL22 fusion protein in infected mice, the expression of IL22R in the lungs, small intestine, colon, liver and spleen was assayed using IHC and RT-qPCR. The results showed that the expression of IL22R was significantly increased in the small intestines and colon both in the normal group and H1N1-infected group (Figure 3A,C). Futhermore, compared with normal group, the expressiong of IL22 in H1N1-infected group was more higher (*p* < 0.05) (Figure 3C). Treatment with vmab-IL22 inhibited the excessive expression of IL22R in the intestines (Figure 3A). Consistently, the secretion of IL22 in intestinal homogenate was significantly decreased under the attack of a virus (*p* < 0.01), compared with the normal group. Treatment with vmab-IL22 significantly enhanced the expression level of IL22 in the small intestine (*p* < 0.001) (Figure 3B). In order to determine the response of the target organs to the recombinant protein, we examined the expression of the IL22 receptor in multiple organs. As shown in Figure 3C, the expression of IL22R was significantly higher in the small intestine and colon, but maintained a lower expression in the lung, spleen and liver in both the normal group and H1N1 group. This suggests that IAV induced secondary immune injury and caused the damage to the intestinal epithelial barrier with an impaired repair ability. Then, the fusion protein treatment specifically increased the expression of IL22 located in the small intestines, which was helpful for promoting the repair of the intestinal barrier. Its efficiency was superior to the mIL22Fc group and vmab + mIL22Fc group, which indicated that the fusion protein exerts more protective activities in the absent of vmab.

### 3.4. Vmab-IL22 Inhibits TLR4 and NLRP3 Inflammasome Activation in the Lung and Intestine in Influenza Virus-Infected Mice

Due to the damaged intestinal barrier, the pathogenic and exposed commensal bacteria activate TLR4 and initiate gut inflammation. At the same time, the NLRP3 inflammasome promotes inflammatory cell recruitment and regulates immune responses in the gastrointestinal tract. The IHC results of TLR4 and NLRP3 in the lungs (Figure 4A,C) and small intestine (Figure 4B,D) was as shown in Figure 4, The number of cells with positive staining increased both in the lungs and small intestine in the H1N1-infected group, and the bifunctional protein significantly inhibited the expression of TLR4 and NLRP3 and exerted anti-inflammatory activity. 

### 3.5. Vmab-IL22 Protects against DSS-Induced Intestinal Injury

IAV infection indirectly causes intestinal immune damage. However, whether the vmab-IL22 fusion protein protects against direct barrier damage remains unknown. Hence, a mouse model of UC was established in male C57BL/6 mice using DSS treatment, and the efficacy of the fusion protein against DSS-induced intestinal barrier damage was evaluated. The results are shown in Figure 5. Compared with the control treatment, 2.5% DSS intake could cause significant intestinal injury, which reflected in the shortened colon and the loss of weight. The fusion protein significantly restored colon length (*p* < 0.01) (Figure 5A,B) and improved body weight (*p* < 0.001) (Figure 5C,D), compared with the DSS control group. Treatment with mIL22Fc and the vmab + mIL22Fc group showed a reversal trend, but without a statistical difference. During the observation process, the vmab treatment in contrast aggravated the intestinal injury.

### 3.6. Vmab-IL22 Strengthens the Intestinal Barrier and Inhibits the Inflammatory Response and Apoptosis in DSS-Treated Mice

The damaged intestinal barrier alters the homeostasis of the gut microbiota. Pathogen-associated molecular patterns and damage-associated molecular patterns are activated by commensal and pathogenic bacteria, activating p-NF-κB p65, NLRP3 to promote inflammatory cell recruitment in the gastrointestinal tract. The inflammasome regulates caspase-1 activation to promote the maturation and secretion of the cytokine precursors pro-IL-1 and pro-IL-18. Meanwhile, the apoptosis in intestinal epithelial cells is driven by increased cytokine activity. Further, when tissue hypoxia occurs and cell permeability increases, cytochrome C can act as a respiratory activator and improve cellular respiration and promote substance metabolism. In our study, the results showed that DSS induced the damaged intestinal barrier and vmab-IL22 improved the integrity of the intestinal barrier (Figure 6A). As shown in Figure 6B, the expression of cytochrome C, cleaved caspase-3, NLRP3, p-NF-κB p65 and cleaved pro-IL-β was significantly increased in the DSS model group due to the increased apoptosis and inflammation. The fusion protein inhibited the expression of cytochrome C, cleaved caspase-3, NLRP3, p-NF-κB p65 and cleaved pro-IL-β, suggesting that it inhibited the inflammatory response and promoted the repair of the intestinal barrier.

## 4. Discussion

A clinical study showed that patients with severe pneumonia present with intestinal barrier dysfunction, which contributes to the pathological process of systemic inflammation and sepsis by increasing intestinal permeability and causing bacterial translocation [27]. Repairing the intestinal barrier is important and helpful for preventing pathogenic bacteria from reaching other parts of the body and causing secondary complications. The results suggested that vmab-IL22 strengthens the intestinal barrier in H1N1-infected mice, which was helpful in reducing lung and intestinal injury by blocking IL17 and supplementing IL22. IL17 is secreted by Th17 and NK cells and acts as an important inflammatory mediator in diseases such as psoriasis, dermatitis and viral pneumonia [28]. Accumulating evidence confirms the complexity of IL-17a bioactivity. It has been reported that the secretion of IL17 significantly increases and is accompanied by higher inflammasome activation in DSS-treated mice. Th17 cells may also have protective functions; the neutralization of IL-17A failed to induce any improvement in Crohn’s disease [29]. IL22 is a member of the IL10 family and has important functions in host defense and tissue repair. IL22 is produced by different types of T cells, including CD4^+^T cells and innate lymphoid cells (ILC3). A previous study demonstrated that mammalian IL22 contributes to host defense at mucosal surfaces and also helps in maintaining the mucosal epithelial barrier via antimicrobial peptide (AMPs) production and anti-apoptotic effects [20]. Elevated levels of IL22 secreted from T lymphocytes were implicated in rheumatoid arthritis and interstitial lung diseases [20]. Exogenous recombinant IL22 protects mice against acute severe pancreatitis-associated lung injury by regulating the apoptotic balance to strengthen the pulmonary microvascular endothelial barrier via the STAT3 signaling pathway [14,30,31].

The interaction of cytokines and the nature of binomial regulation leads to functional complexity. The specific role of cytokines depends on their microenvironment, and this has been confirmed in several studies. In a model of bleomycin-induced pneumonia, IL22 was shown to lose its protective effect and instead promoted airway inflammation in the presence of IL17A. However, its tissue-protective activities were strengthened in the absence of IL17A. This suggested that the presence of IL17A in the microenvironment determines whether IL22 acts as a pro-inflammatory or tissue-protective factor [24]. The increase in IL17 and IL22 in Crohn’s disease is associated with disrupted epithelial regeneration and limits the expansion of intestinal stem cells (ISCs) [15]. The damaged mucosal epithelium could trigger the activation of many related immune cells to secret IL22, which contribute to maintaining the mucosal epithelial barrier. Hence, it is important to understand how the balance of cytokines can be modified in favor of intestinal barrier repair.

In order to amplify the advantages of previous biological drugs, our group constructed a fusion protein by combining an anti-IL17 antibody with mIL22 [26]. The fusion protein could reduce inflammation by neutralizing IL17 and strengthening the repair function of IL22. We established an acute viral pneumonia model with indirect gut injury induced by IAV (H1N1), and the efficacy of vmab-mIL22 against gut immune injury was examined. Treatment with vmab-mIL22 effectively alleviated pulmonary edema and congestion but also strengthened the intestinal barrier when compared with treatment with the combination of vmab and IL22Fc and vmab or IL22Fc alone. Further, our results showed vmab-mIL22 inhibited the TLR4/NLPR3 signaling pathway and upregulated the expression of IL22R in the small intestine, which promoted repair and inhibited the inflammatory cascade reaction initiated by H1N1. Notably, we compared the expression of IL22R in the lungs, small intestine, colon, liver and spleen tissue between the normal and model groups. The results showed the expression of IL22R was highest in the small intestine and colon. After infection, the expression of IL22R became higher in the small intestine tissue than in the normal group. The small intestine was the target organ of the fusion protein, which was better for intestinal repair. Hence, the increase in IL22R served as a protective immune defense against intestinal injury after viral infection, although it was insufficient to reverse the trend of intestinal injury. 

Our previous study demonstrated the potential efficacy of the fusion protein against lung injury in H1N1-infected mice [26]. Therefore, this fusion protein can effectively intervene against intestinal immune injury, and indirectly contribute to treat lung injury in viral pneumonia. The present study focused on the effect of this protein against intestinal injury in the same model. Since vmab-mIL22 protects against both lung and small intestine injury, we examined its direct effect on gut injury so as to clarify the therapeutic value of the fusion protein in cases of both indirect immune injury and direct injury. A DSS-induced enteritis model was used to examine the direct protective effect of vmab-mIL22 in the intestine. We confirmed its protective effect against the damaged intestinal barrier caused by DSS, which was related to the inhibition of inflammatory pathways, such as cytochrome C, caspase-3, IL-β, NLRP3 and NF-kappa B, and we found that the recombinant protein promoted repair by inhibiting apoptosis and exerting antioxidative effects.

This study preliminarily evaluated the efficacy of the fusion protein vmab-mIL22 on the gut immune injury induced by the H1N1 virus and gut injury caused by the DSS treatment. Our results showed that vmab-mIL22 had potent synergistic anti-inflammatory and repair-promoting activities through blocking the IL17A pathway and activating the IL22 pathway in an influenza and colitis model. In view of the significant efficacy of Vunakizumab-IL22 against immune and no-immune enteritis, we believe that the fusion protein could be a potential biomedicine for this condition. There are still some limitations in our study; the female mice were not fully considered in the experiment design. In addition, the long-term safety of Vunakizumab-IL22 has not been evaluated. and how the fusion protein affects the crosstalk between the lungs and gut, and whether it affects the function and number of common immune cells in the lungs and gut, which need further investigation.

In conclusion, Vunakizumab-IL22 could act as a novel therapeutic agent against inflammatory mucosa damage occurring due to the immune injury induced by the influenza virus and intestinal bowel disease. The therapeutic strategies to improve the intestinal barrier and promote repair would prevent the pathological process of severe viral pneumonia.

## Figures and Tables

**Figure 1 biomedicines-11-01160-f001:**
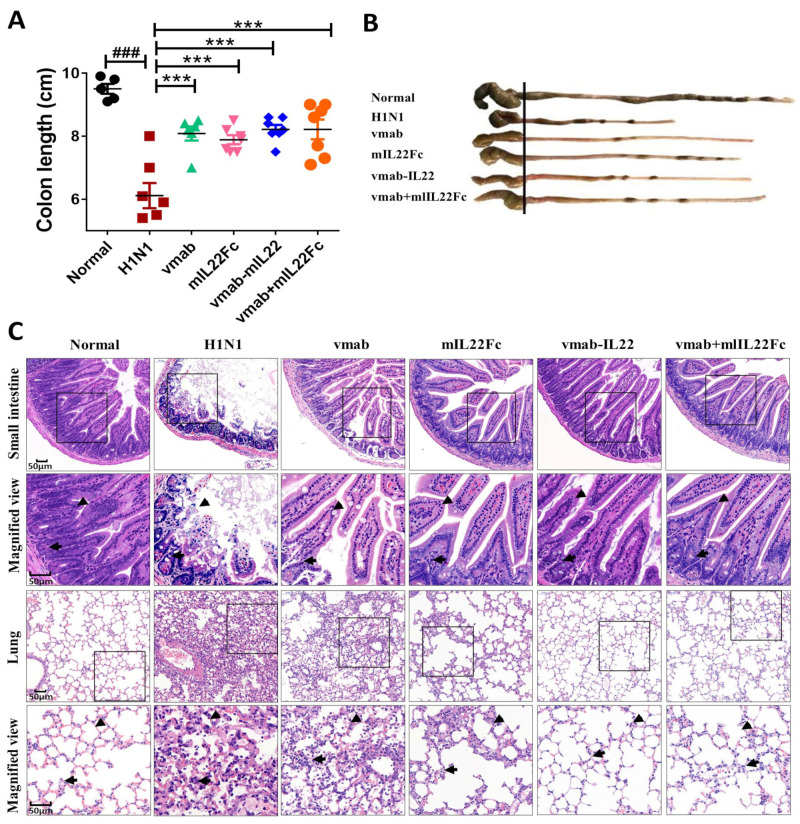
Vmab-mIL22 treatment protects against the lung and gut injury induced by the influenza A virus in mice. Four-week-old BALB/c mice were infected intranasally with the influenza virus at a dose of 2LD_50_/mouse in 30 μL DMEM (*n* = 6 for each group). Treatment was administered intravenously at 2.5 h and 48 h after infection. Mice were sacrificed at 5 days in both groups. (**A**) The length of the colon at day 5 in H1N1-infected mice. (**B**) Representative images of the colon at day 5 in H1N1-infected mice (**C**). Pathological changes in the small intestine and lung were evaluated using H&E staining. Intestinal villi morphology (triangle) and intestinal inflammatory infiltration (arrow) in above; alveolar morphology (triangle) and lung inflammatory cell infiltration (arrow) in below. ### *p* < 0.001 for comparison of mice in H1N1 group to the normal group, *** *p* < 0.001 for comparison of mice in treated group to the H1N1 group.

**Figure 2 biomedicines-11-01160-f002:**
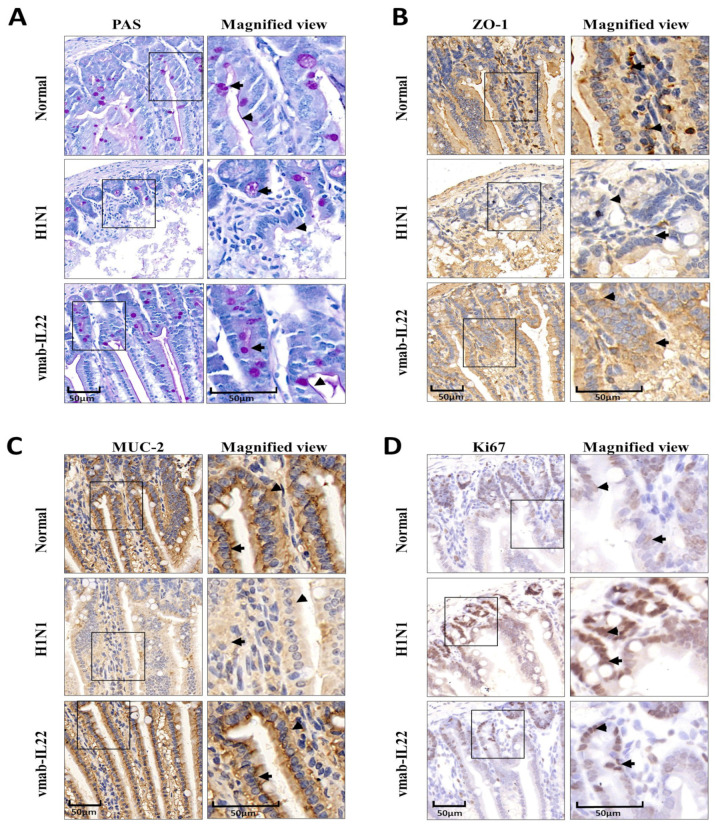
Vmab-mIL22 treatment improves intestinal epithelial repair and promotes regeneration. Male BALB/c mice were infected with the H1N1 virus and then intravenously treated with vmab-mIL22 twice in 4 days. Positive cells are indicated by triangles and arrows. (**A**) Staining for PAS, (**B**) ZO-1 expression, (**C**) MUC-2 expression and (**D**) the expression of Ki-67 protein in the small intestine were measured using IHC (*n* = 4). PAS, Periodic Acid-Schiff stain; ZO-1, zonula occludens protein 1; MUC-2, Mucin-2.

**Figure 3 biomedicines-11-01160-f003:**
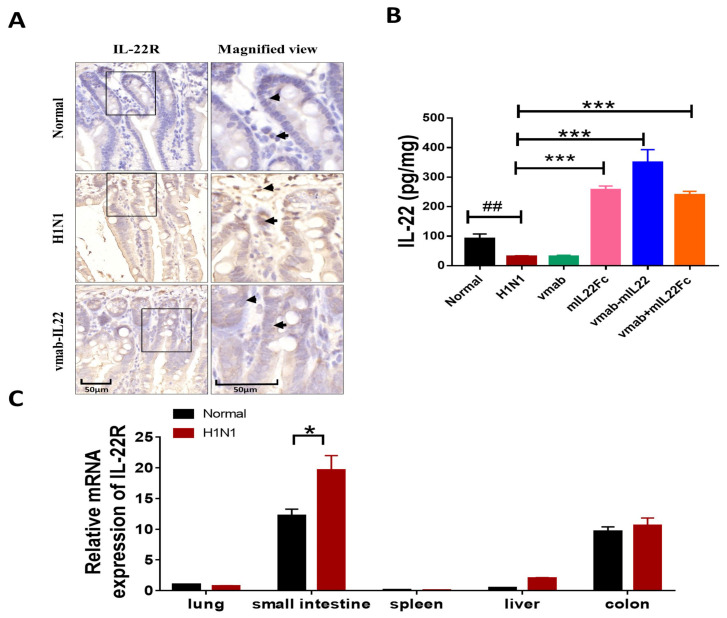
Vmab-mIL22 treatment increases the expression of IL22R in influenza A virus-infected mice. The infection and treatment schemes were the same as in Figure 1. (**A**) The expression of IL22R protein in small intestine was measured using IHC in H1N1-infected mice (*n* = 4); positive cells are indicated by triangles and arrows. (**B**) The expression of IL22 in the small intestine homogenate was assayed using ELISA following treatment with vmab-mIL22 (*n* = 6). (**C**) The distribution and transcription levels of IL22R was determined in the lung, small intestine, colon, liver and spleen using RT-PCR in H1N1-infected mice (*n* = 4).## *p* < 0.01 for comparison of mice in H1N1 group to the normal group, * *p* < 0.05, *** *p* < 0.001 for comparison of mice in treated group to the H1N1 group.

**Figure 4 biomedicines-11-01160-f004:**
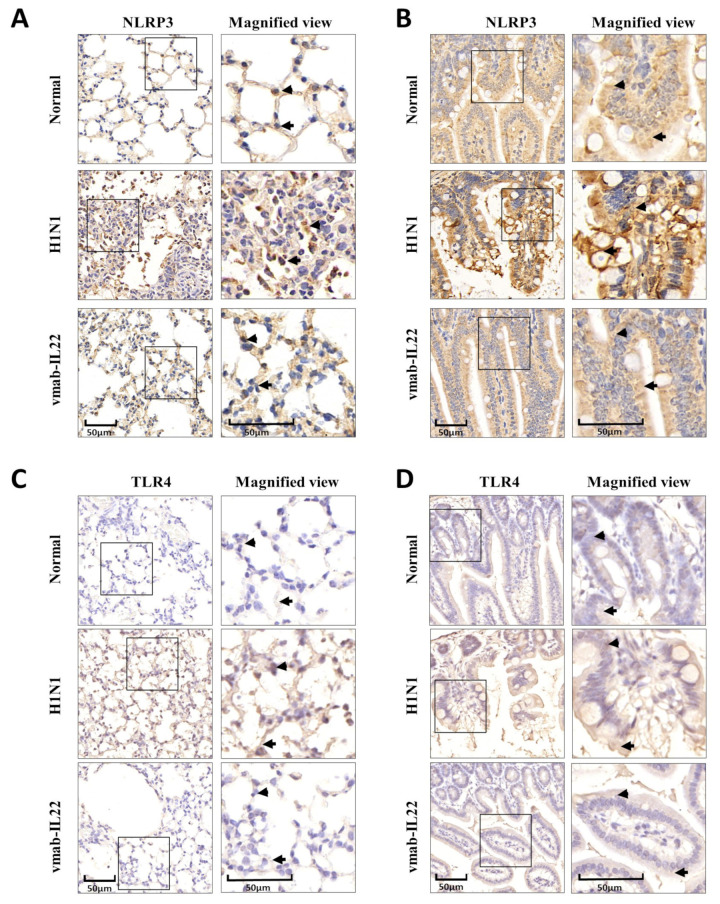
Vmab-mIL22 treatment inhibits the expression of NLRP3 and TLR4 in the lung and small intestine in influenza A virus-infected mice. The infection and treatment schemes were the same as in Figure 1. The expression of NLRP3 (**A**,**B**) and TLR4 (**C**,**D**) was assayed using IHC in the lungs and small intestine (*n* = 4). Positive cells are indicated by triangles and arrows. NLRP3, NOD-like receptor thermal protein domain associated protein 3; TLR4, toll-like receptor 4.

**Figure 5 biomedicines-11-01160-f005:**
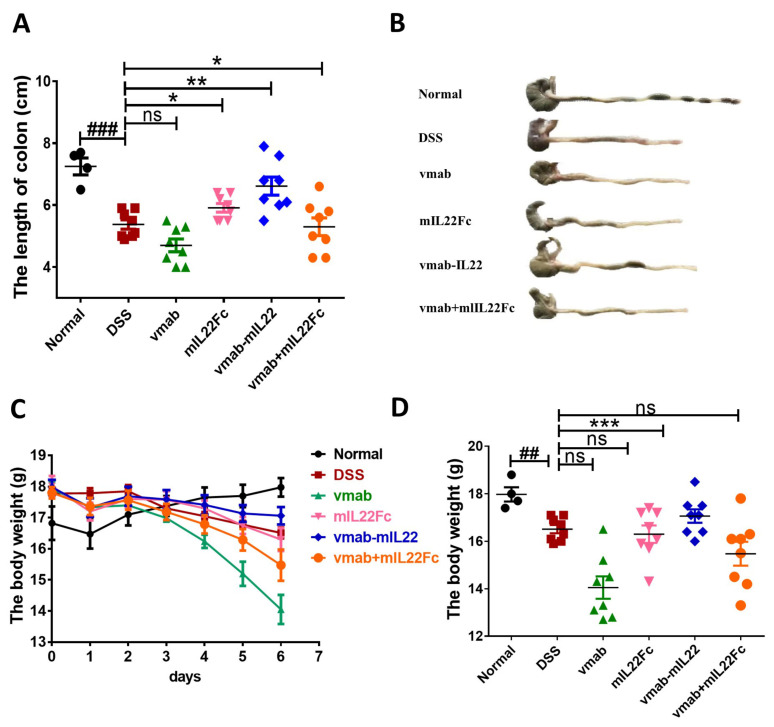
Vmab-mIL22 treatment reduces the damage to the colon induced by DSS in mice. Male C57BL/6 mice were treated with 2.5% DSS for 7 days, which was followed by the intravenous administration of 10.66 mg/kg vmab-mIL22 twice in 10 days. (**A**) The length of the colon at day 7 in DSS-induced colitis mice (*n* = 8 for each group); (**B**) representative images of the colon at day 7 in DSS-induced colitis mice; (**C**) change in weight in DSS-induced colitis mice during the 7 days; (**D**) change in weight in DSS-induced colitis mice at day 7. ### *p* < 0.001, ## *p* < 0.01 for comparison of mice in H1N1 group to the normal group, *** *p* < 0.001, ** *p* < 0.01, * *p* < 0.05, for comparison of mice in treated group to the H1N1 group.

**Figure 6 biomedicines-11-01160-f006:**
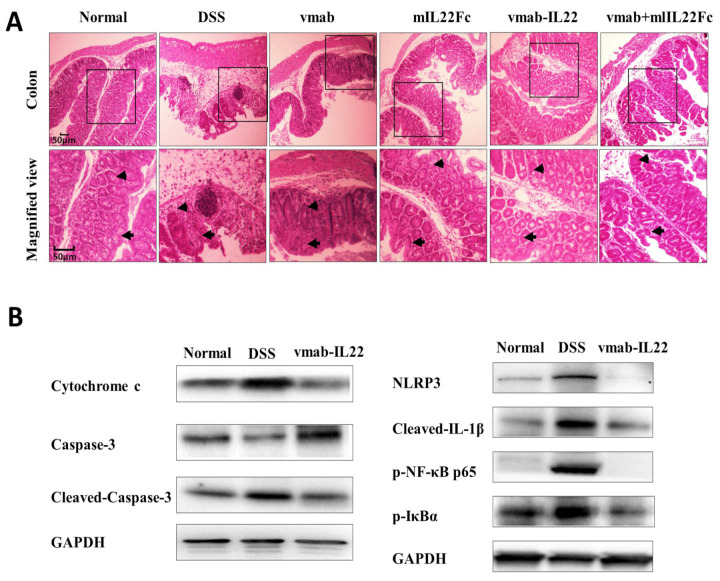
Vmab-mIL22 treatment attenuates the damage to the intestinal barrier and regulates apoptosis and inflammatory pathways in DSS-induced mice. (**A**) Pathological changes in the colon were evaluated using H&E staining. Intestinal villi morphology (arrow) and intestinal inflammatory infiltration (triangle) were indicated. (**B**) Western blot analysis of cytochrome c, caspase-3, cleaved caspase-3, NLRP3, cleaved-IL-1β, p-NF-κB p65 and p-IκB-α in expression in the colon in DSS-induced colitis mice. NLRP3, NOD-like receptor thermal protein domain associated protein 3; p-NF-κB p65, phospho nuclear factor-kappa B; p-IκBα, phospho inhibitor of nuclear factor (NF)-κB-α; GAPDH, glyceraldehyde-3-phosphate dehydrogenase.

**Table 1 biomedicines-11-01160-t001:** Primer sequences used for RT-qPCR.

Gene Name	Primer Sequences
GAPDH	Forward primer: 5′-AGGTCGGTGTGAACGGATTTG-3′; Reverse rimer:5′-TGTAGACCATGTAGTTGAGGTCA-3′;
IL22R	Forward primer: 5′-TACGTGTG CCGAGTGAAGAC-3′; Reverse primer: 5′-TAACAGAGCAAGCCGACGAG-3′

## Data Availability

The data will be available through the corresponding author upon request.

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
