# Peer review of "Vunakizumab-IL22, a Novel Fusion Protein, Promotes Intestinal Epithelial Repair and Protects against Gut Injury Induced by the Influenza Virus"

_biomedicines, 2023, doi:10.3390/biomedicines11041160_

Round 1
Reviewer 1 Report
The study used mouse models to test the treatment of the effects of H1N1 infection on the lung and intestinal mucosa using a recombinant IL17A-IL22 antibody (vmab-IL22). The effects of infection and treatment combinations were analyzed macroscopically, and histopathologically, by the number of goblet cells, the protein expression of cell-associated proteins and infection-induced systems, and by the maturation of cytokine levels.
It was found that vmab-IL22 protects against lung and intestinal injury induced by H1N1 infection, strengthens the intestinal barrier, promotes the intestinal expression of IL22R, inhibits pulmonary and intestinal TLR4 and NLRP3 inflammasome activation, and ameliorates DSS-colitis by strengthening the intestinal barrier.
The experiment is well thought out, logically designed, and well-documented. The fusion protein generation is clearly described. The results are valid, and the measurements and the macro- and microscopic examinations are well documented. Figures and diagrams aid in understanding the results.
The results provide new evidence for a treatment strategy for severe viral pneumonia involving intestinal barrier defense. Indeed, the results suggest that vmab-IL22 is a promising biopharmaceutical and a candidate for the treatment of direct and indirect intestinal injuries, including those induced by the influenza virus and DSS.
The use of English is appropriate; minor polishing is required.
Once this has been done, I recommend accepting the article for publication.
Reviewer 2 Report
The manuscript shows the results of studies aimed at investigating the effect of a fusion protein Vunakizumab-IL22 on the parameters of inflammation and gut permeability in mice experimentally infected with the H1N1 virus. The studies are in general well designed and clearly described, the inclusion of a DSS-treated group is a great advantage of the work.
Unfortunately, the manuscript does not meet the formal requirements of the journal regarding original images for blots, which state: “In order to ensure the integrity and scientific validity of blots (including, but not limited to, Western blots) and the reporting of gel data, original, uncropped and unadjusted images should be uploaded as Supporting Information files at the time of initial submission.” All the blots submitted as supplementary material are cropped, therefore Authors should provide original, unprocessed images before acceptance for publishing.
At the end of the Discussion section, the Authors should add a comment on further studies that should be carried out to prove the efficacy of the Vunakizumab-IL22 before its use as a therapeutic agent in humans.
It would be very useful for readers to indicate on histological images (e.g. with arrows) the elements of interest.
Detailed comments
Line 30 H1N1
Line 34 please correct „clone” to “colon” in the whole manuscript
Line 104 “Details of the construction, expression, and purification of this fusion protein are provided in our previous paper[2].” The reference should be revised as it is not the Authors’ paper. Please, revise also all other references.
Line 149 Please explain “CMC Na” at the first use
Section 2.9 – please add information on checking parametric and nonparametric statistical test parameters (distribution of variables and homogeneity of variances), which should be performed before the selection of statistical tests.
Line 227 please add references
Line 240-241 please provide a reference for the statement
Line 361 please correct “infection (p<0.01), compared”
Line 553-556 please provide a reference for the statement
Line 587-590 “In order to amplify the advantages of previous biological drugs, our group constructed a fusion protein by combining an anti-IL17 antibody with mIL22. The fusion protein could reduce inflammation by neutralizing IL17 and strengthening the repair function of IL22.” Please provide a reference or clearly state if this statement concerns the present work.
Line 605-606 please provide a reference
Line 618 please correct “DSS treatment”
Reviewer 3 Report
The authors performed an elegant study (another brilliant idea, after their first discovery), resulting in a very good manuscript, with potential practical importance also in humans. I have only minor comments, listed below:
1. Abstract: Please rephrase the aim, so that it appears complete: “In this study, we investigated that the protective effects against enteritis given its anti-inflammatory and tissue repair functions.” (or just delete “that”).
2. Keywords: I suggest “Dextran sulfate sodium” to be added here, as it does not belong to the title and important effects were proven with Vunakizumab-IL22 in mice with DSS-induced colitis. Also, since it is not mentioned in the title, “mice” could also be added to Keywords. The importance of Keywords is to improve indexing.
3. Introduction: Please include only the aim of your study (by the end if Introduction) and nothing referring to results / findings. This is not a summary of the paper.
4. Materials and Methods:
* Line 104 – reference [2] refers to other paper, not that of the authors. Please correct.
* Lines 141, 193, 358, 406 – please correct “clone”.
5. Results:
* Lines 248-250. I suggest to remove “The results suggested that vmab-IL22 was helpful in reducing lung and intestinal injury by blocking IL17 and supplementing IL22”, as it could be developed in “Discussion.
* Also, please delete “D” from Figure 1, as the main text and the figure legend explains it well (1C).
* Line 358 – please correct “evaluated”.
* Figure 4 – please correct figure legend “The expression of NLRP3 (A) and TLR4 (B) was assayed using IHC in the lungs and small intestine (n=4).” – B should be replaced by “C” and also explain the present “C and D”. And, also please add in the main text.
* Please insert place for Fig. 5B and Fig. 5C in the main text.
* Lines 474-476: I suggest to remove “The results suggested that the anti-IL17A/IL22 fusion protein protects against the direct intestinal injury caused by DSS. In the absence of IL-17A, IL22 exert more effective on protection of intestinal barriers” from here and insert them in “Discussion”.
6. Discussion:
* I suggest that some of the sentences (at the beginning of Discussion) regarding IL-17 and IL-22 to be moved into “Introduction”, as they represent general data. Please start Discussion with the findings of your study and then expand.
* Please insert limitations of your study. I know that female mice have not been widely used, as there were considered issues regarding hormones. However, they started to be used. Using only males would introduce a bias. This should be discussed as a limitation of the study. See paper by Smarr et al, 2022, https://doi.org/10.1186/s13293-022-00451-1.
* Line 631 – please replace “intestinal” with “inflammatory”
* Please provide proper future directions for research.
7. Please also provide the conclusion by the end of Discussion. The authors concluded their findings, therefore just insert “in Conclusion” or just “Conclusion”.
Round 2
Reviewer 2 Report
I accept the responses of Authors to my comments and accept the manuscript for publication.